# Evaluation of airflow pattern and thermal behavior of the arched greenhouses with designed roof ventilation scenarios using CFD simulation

He Li[1,2], Yiming Li[1,2], Xiang Yue[1,2], Xingan Liu🔘[2,3]*, Subo Tian[1]*, Tianlai Li[2,3]

**1** College of Engineering, Shenyang Agricultural University, Shenyang, China, **2** National & Local Joint Engineering Research Center of Northern Horticultural Facilities Design & Application Technology (Liaoning), Shenyang, China, **3** College of Horticulture, Shenyang Agricultural University, Shenyang, China

\* lxa10157@syau.edu.cn (XL); tiansubo@syau.edu.cn (ST)

**Data Availability Statement:** All relevant data are within the manuscript and its Supporting Information files.

## Abstract

Natural ventilation is an effective energy-saving strategy conducive to promoting sustainable agricultural production. A comprehensive numerical simulation is performed to predict the airflow pattern and thermal behavior in different arched greenhouses. The defined arc chord angle and position angle are employed to examine the natural ventilation process and corresponding roof vent scenarios. The numerical simulation is compared with the experimental data and good agreements are observed. Various configurations of ventilated structures, wind conditions and ventilation layouts are simulated on a high-resolution polyhedral grid based on a grid sensitivity analysis, which is beneficial to the optimization of greenhouse cooling combined with the water circulation heat collection system. The cooling effect in summer is analyzed by estimating the ventilation flow rate and microclimate inhomogeneity. The results demonstrate that the position angle of 85˚ of the arched greenhouses is an optimum ventilation direction and its impact on the microclimate is marginally affected by the change of the ventilation structure. The designed ventilation scheme has perfect air exchange capacity and cooling effect because the average air temperature can be reduced by 1.5˚C more than the existing greenhouse in 10 minutes of ventilation. Likewise, the results show that the temperature and velocity inhomogeneities are approximately decreased by 33.3% and 11.89%, respectively. The practical value of the research is expected to provide basic quantitative conclusions for evaluating the natural ventilation performance.

## Introduction

Greenhouse production is an effective strategy for the intensification of modern agriculture, providing the suitable microclimate for crops while energy consumption and environmental pollution continue to increase [1, 2]. It has guaranteed vegetable supplies and contributed to raise national income due to the rational use of solar energy. Environmental parameter

**Funding:** The present work is financially supported by the Postdoctoral Foundation of Xingan Liu under the Grant 1105/770218004. Xingan Liu had a role in the study design, data collection and analysis, decision to publish, or preparation of the manuscript.

**Competing interests:** The authors have declared that no competing interests exist.

regulation of the greenhouse is critical to prevent problems that the excessively high air temperature is often appeared in summer season [3, 4]. Therefore, the airflow pattern and thermal behavior of the greenhouse are considered the great important design feature so that transpiration and photosynthesis of crops can be maintained at satisfactory levels. Natural ventilation is the most effective and economical measure to control greenhouse climate since it directly realizes the transport of heat and mass between the internal and external environment under the premise of energy saving [5–7]. The efficiency of natural ventilation can be used to control variables such as temperature, carbon dioxide concentration, humidity [8, 9]. Its driving force is the combination of wind and buoyancy forces [10]. Frequently, it is strongly influenced by greenhouse ventilation scenarios (i.e. vent structure, vent position, vent opening) and the external wind characteristics (i.e. wind speed, wind direction). From the architecture point of view, adjusting the vent configurations significantly impacts the ventilation efficiency and spatial heterogeneity of the airflow [11, 12]. In Northeast China the sliding cover solar greenhouse has been developed, with skeleton structure is a semi-circular arc, and it is characterized by bitter winter energy capture and less summer cooling load for the production of crops [13]. However, the excessively high internal air temperature is often generated due to the unreasonable ventilation configuration of the greenhouse. Moreover, there is a lack of relevant research that specifically designs the roof ventilation structure and vents layout.

Several studies have been carried out to analyze and optimize the natural ventilation process, but most of them have focused on multi-span greenhouses rather than single-span greenhouses. Bournet et al. [14] performed a computational fluid dynamics simulation to predict the effect of venting arrangements on ventilation and microclimate parameters in pitched roof glasshouse. Villagrán et al. [15] investigated the influence of different structures of the roof vent on the climate distribution of multi-span greenhouses to propose the recommended greenhouse applying in the high Andean tropics. Kim et al. [16] developed numerical model to evaluate wind pressure coefficients of the various ventilation variables used for greenhouse design of reclaimed coastal land such as roof slope, roof curvature radius and number of spans. K. He et al. [17] simulated the cooling and dehumidification processes inside a 11-span ventilated plastic greenhouse with different vent configurations. They revealed that the roof and side openings have the best cooling and homogeneity performance. Furthermore, greenhouse climatic parameters and standard deviations are less influenced by span number than vent configuration [18]. In addition, the study of natural ventilation and the microclimate in different types of greenhouse has been approached, and there are few parameterized summary analyses of the arched greenhouses, especially the roof ventilation configurations. Ould Khaoua et al. [19] and Bartzanas et al. [20] assessed the ventilated performance of vent forms (i.e. The roll-up vents, the pivoting vents), considering the variation of crop aerodynamic resistance and climate homogeneity. Kittas and Bartzanas [21] analyzed the impacts of two ventilation openings on the dehumidification process. They concluded that the latent heat and sensible heat exchange rate of pivoting door type are higher than that of the roll-up type. Zhang et al. [22] investigated the impacts of the various vent openings on the air humidity in Chinese solar greenhouse. Therefore, optimization modelling of the ventilated arched greenhouse should be analyzed quantitatively considering the vent structure and geometrical parameter.

Several methodologies of the natural ventilation have been fully developed (i.e. field experiments, wind tunnel tests and numerical simulation). Field tests can only analyze the airflow state at the local position and it is difficult to control the interference of various uncertain factors. Some experimental studies have actually produced satisfactory results [23, 24]. López et al. [25] demonstrated that roof vents designed to open towards the leeward, generate a positive interaction between the wind and stack effects, can improve the ventilation efficiency of greenhouses. Additionally, López et al. [25] used sonic anemometry to assess the effect of the

number of half-arch roof vents on airflow pattern of the three-span Mediterranean greenhouse. This ventilator configuration takes into consideration the obstruction caused by the surrounding greenhouses and air movement at the plant levels. However, it is unreasonable to use the purely experimental method to assess the parameterization of various greenhouse ventilation structures. To overcome the limitations of cost and labor consumption problems, computational fluid dynamics (CFD) methodology has been used in the present study to provide detailed information about the distribution of microclimates. This simulation method has made it possible to accurately evaluate and interpret ventilated condition within a virtual environment [26–30]. Benni et al. [11] conducted an optimization configuration for the roof and side vents of greenhouse natural ventilation. They found a closed windward roof can remove 64% of the maximum heat. Pakari and Ghani [31] utilized the CFD simulation to analyze the air change rate and airflow distribution on a greenhouse equipped with wind towers. He et al. [32] established the ventilated model to optimize the vent dimensions of back walls and the results derived from simulation data have been proven to be reliable. The CFD simulation is an essential tool for the understanding of internal airflow patterns and temperature profiles, which benefits engineer and greenhouse manufacturers to improve the greenhouse design [33, 34].

The objective of this study is to optimize ventilation structures of arched greenhouses adopting the CFD simulation according to the airflow pattern and thermal behavior. The simulation modeling is implemented through the existing arched greenhouse in combination with the reference geometry of the sliding cover solar greenhouse. Further, to reinforce the credibility of the predictive model of roof ventilation, the numerical model has been validated against experimental data. The performance of three ventilation structures of Rolling shutter type (RS), Jack-up type (JU) and Pivoting window type (PW) are evaluated. Moreover, a parameterized investigation is presented by the geometric feature of the roof ventilation configuration with the reasonable arc chord angle and the position angle for maximum ventilation efficiency. The cooling effect in summer is analyzed by estimating the ventilation flow rate and microclimate inhomogeneity. Finally, the most suitable roof-vent design scheme is selected, and conclusions are made on the optimal design of roof ventilation for arched greenhouses.

## Materials and methods

### Description of the arched greenhouse

The arched greenhouse with assemble skeleton structure made of galvanized steel pipes, with a length of 60 m, width of 10.4 m, roof angle of 45° and maximum net production area of 540 m$^2$, The experimental greenhouse is located in the Horticulture Facility Design & Environmental Control Research Institute of Shenyang Agriculture University, China (latitude: 41.8°N, longitude: 123.4°E, altitude: 42 m). The arc length of the fixed thermal insulation layer is 5.6 m on the north side of the greenhouse, which the thermal insulation material consists of 75 mm thickness polyurethane benzene plates. The outside of the south roof adopts rock wool color steel plate sliding cover for heat preservation, which the sliding cover is filled with glass wool carpets. The east and west gables is established with movable thermal insulation. The transparent covering material of the south roof is 0.2 mm thickness polyolefin film. The thermal insulation layer on the north side of the greenhouse is equipped with a water circulation heat collection system, which can adjust the heat storage and release to replace the back wall of the traditional Chinese solar greenhouse. During the summer, the water source of coming from the underground water. It relies on the circulation of cool underground water to absorb excess heat, thereby optimizing the operating temperature of the greenhouse [35]. The operation time of the system is 10:00–15:30 and the average water temperature in the water channel is about 17°C. The continue roof vent is equipped a pivoting type ventilation structure with

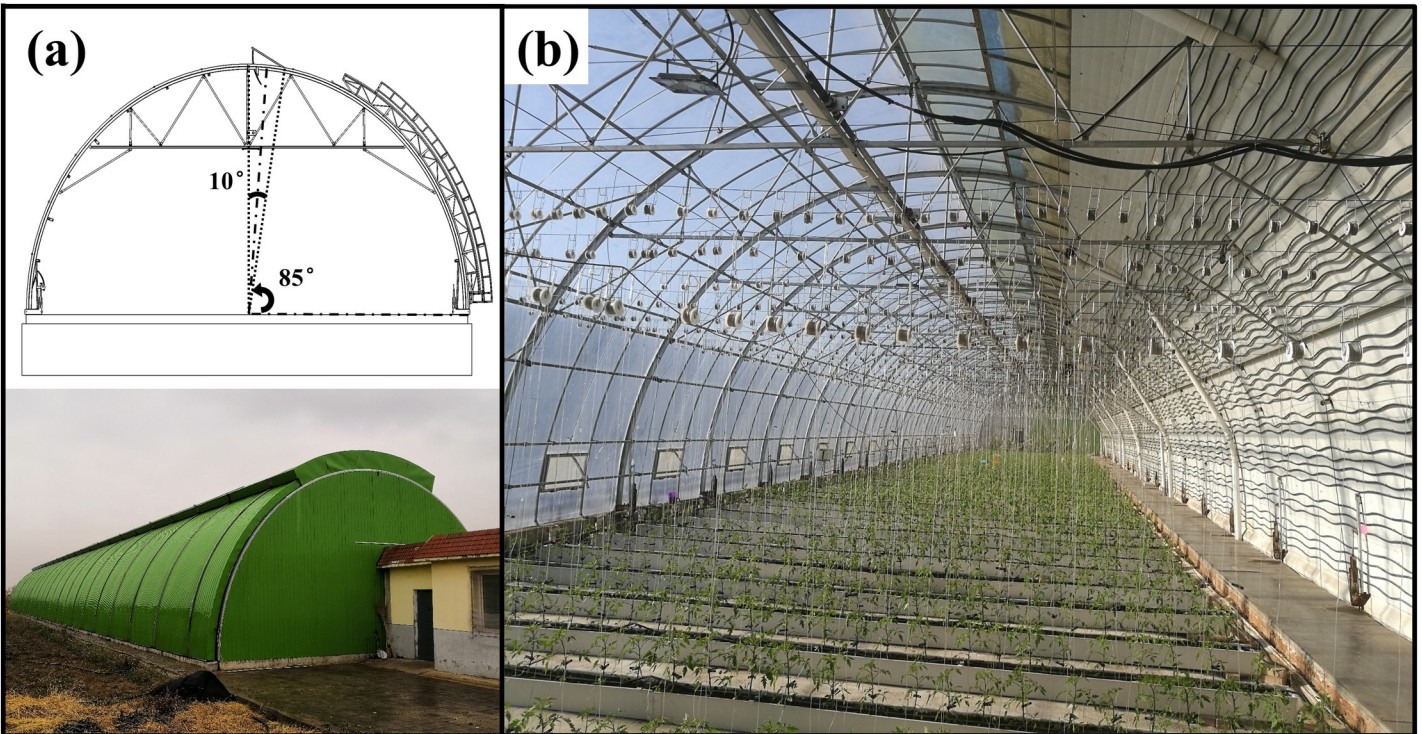

**Fig 1.** (a) Skeleton structure views of the arched greenhouses with continuous roof ventilation. (b) Inside view of the greenhouse equipped with a water circulation heat collection system.

the arc chord angle of 10° and the position angle of 85° (as shown in **Fig 1**). During the experiment, in order to systematically study the ventilation configuration of roof vent, the ventilation at the bottom corner of the southern roof was always closed and the influence of crop growth conditions on the ventilation configuration was ignored.

## Measurements and experimental procedures

An experimental setup was performed for validation of the numerical model and the experimental data were recorded to improve the accuracy of the boundary conditions of the simulation model. Thermocouples (TYD-ZS2 type, range: -40 to 80°C, precision: ±0.2°C, resolution: 0.1°C) coupled to a data logger (HOBO U100-011, USA Oneset Co Ltd) were measured air temperature at locations shown in **Fig 2**. The collected trial data were recorded at minute intervals and eventually output in Excel format. Meanwhile, the hot-wire anemometer (TSI-9535, range: 0–30 m s$^{-1}$, precision: 0.01 m s$^{-1}$, accuracy: ±3%) was used to measure the airflow through the roof opening of the arched greenhouse. The measured data were collected with constant time step of 30 s. External conditions were monitored though web crawler technology. Air temperature, weather sunny factor, wind speed and wind direction were recorded every 30 min. These environmental parameters were adopted as the boundary conditions for the numerical simulation. Moreover, the predicted root mean square division (PRMSD) and normalized mean square error (NMSE) of the numerical models are calculated by **Eqs 1 and 2** to thoroughly verify the overall performance.

$$PRMSD = \frac{1}{D_m}\sqrt{\frac{\sum_1^N (D_m - D_s)^2}{N}} \qquad (1)$$

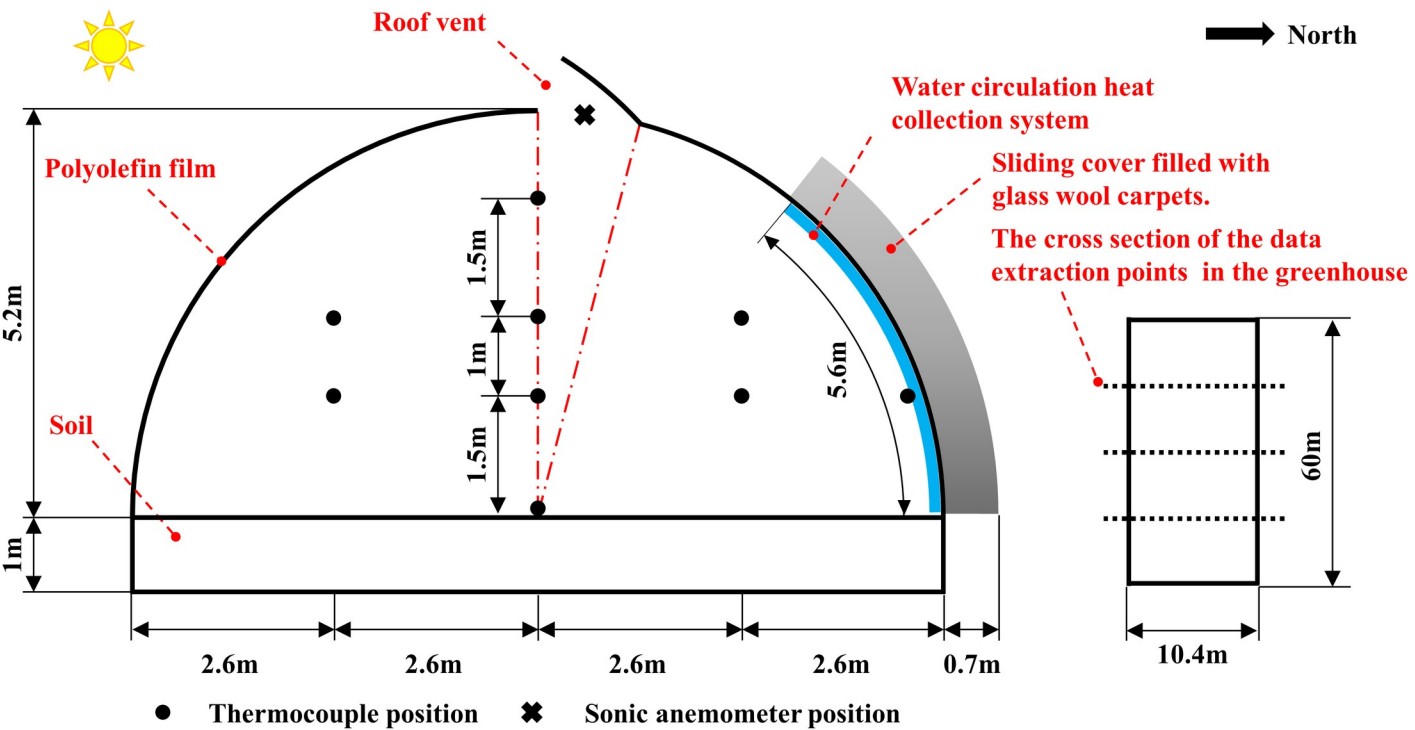

**Fig 2. Schematic diagram of experimental data collection of the arched greenhouse.**

$$NMSE = \frac{\sum_{1}^{N} (D_m - D_s)^2}{\sum_{1}^{N} (D_m D_s)} \qquad (2)$$

where $N$ is the number of measurement points, $D_m$ is the measured parameter, and $D_s$ is the simulated value.

## Modeling and numerical simulation

The CFD simulation easily allows parametric study to analyze hypothetic ventilation configurations, especially when ensure that different configurations are prior embedded in the same computational dominant and grid [35]. It numerically solves a set of partial differential equations for the conservation of mass, momentum, and energy. This paper presents detailed three-dimensional simulations of natural ventilation in arched greenhouses in relation to the designed roof ventilation scenarios with ANSYS 19.2 commercial CFD software. In addition, The geometry and meshes for the numerical analysis are designed using the Solidworks and Fluent Meshing software tools.

**Fundamental equations and physical models.** The established models based on the finite volume method generally solve a set of transport conservation equations for mass (**Eq 3**), momentum (**Eq 4**), and energy (**Eq 5**). The Navier-Stokes equations are solved in combination with the realizable k-ε turbulence model. The turbulence model is embedded to provide accurate airflow field distribution compared to the standard k-ε turbulence model [36]. Consequently, the transport equations for the turbulent kinetic energy and dissipation rate are

modelled by **Eqs 6 and 7**.

$$\frac{\partial \rho}{\partial t} + \nabla(\rho \overrightarrow{v}) = 0 \tag{3}$$

where $\rho$ is the density, $t$ represents the time, and $\overrightarrow{v}$ is the velocity vector.

$$\frac{\partial}{\partial t}(\rho \overrightarrow{v}) + \nabla(\rho \overrightarrow{v} \overrightarrow{v}) = -\nabla P + \nabla \cdot (\overline{\overline{\tau}}) + \rho \overrightarrow{g} \tag{4}$$

where $P$ is the pressure, $\overline{\overline{\tau}}$ is the stress tensor, and $\overrightarrow{g}$ is the gravity acceleration.

$$\frac{\partial}{\partial t}(\rho E) + \nabla \cdot (\overrightarrow{v}(\rho E + P)) = \nabla \cdot (k_{eff} \nabla T - \sum_j h_j \overrightarrow{J}_j + (\overline{\overline{\tau}}_{eff} \cdot \overrightarrow{v})) \tag{5}$$

where $E$ is the flow energy, $k_{eff}$ is the effective conductivity, $T$ represents is the temperature, $h_j$ is the sensible enthalpy, $\overline{\overline{\tau}}_{eff}$ is the effective viscosity shear and $\overrightarrow{J}_j$ is the diffusion flux of species.

$$\frac{\partial}{\partial t}(\rho k) + \frac{\partial}{\partial x_i}(\rho k u_i) = \frac{\partial}{\partial x_j}\left[\left(\mu + \frac{\mu_t}{\sigma_k}\right)\frac{\partial k}{\partial x_j}\right] + G_k - \rho \varepsilon \tag{6}$$

$$\frac{\partial}{\partial t}(\rho \varepsilon) + \frac{\partial}{\partial x_i}(\rho \varepsilon u_i) = \frac{\partial}{\partial x_j}\left[(\mu + \frac{\mu_t}{\sigma_\varepsilon})\frac{\partial \varepsilon}{\partial x_j}\right] + C_{1\varepsilon}\frac{\varepsilon}{k}G_k - C_{2\varepsilon}\rho\frac{\varepsilon^2}{k} \tag{7}$$

where $C_1 = \max(0.43, \eta/\eta+5)$ $\eta = (2E_{ij}, E_{ij})^{0.5}k/\varepsilon$ $E_{ij} = 0.5(\partial u_i/\partial x_j + \partial u_j/\partial x_i)$ $\mu_t = \rho C\mu k^2/\varepsilon$, $k$ is the turbulence kinetic energy and $\varepsilon$ is the dissipation rate of turbulence kinetic energy, $u_i$ is the velocity vector in Cartesian coordinates, $\mu_t$ is the eddy viscosity, $\sigma_k$ and $\sigma_\varepsilon$ are the Turbulent Prandtl numbers, $C_{1\varepsilon}$ and $C_{2\varepsilon}$ are the model constants, $G_k$ is the generation of $k$ due to mean velocity gradients, $G_b$ is the generation of $k$ due to buoyancy, $Y_M$ is the contribution of fluctuating dilation in compressible turbulence to overall dissipation rate.

The solar ray tracing model is applied to the simulation using Discrete Ordinates (DO) due to its advantage of ability for the solution of radiation in the semi-transparent walls, especially span the entire range of the optical thickness [28, 37]. The DO model is not constrained by optical thicknesses and the distribution of luminance is obtained by solving the radiative transfer equation (**Eq 8**). In the solar ray tracing model, geographical location and time zone, model orientation, sunshine factor and simulation time are imported to the model. In order to control the radiation update frequency during continuous phase solution proceeds, the energy iteration per radiation parameter is set to 10 default and illumination parameters enable the solar-calculator.

$$\nabla(I_\lambda(\overrightarrow{r}, \overrightarrow{s})\overrightarrow{s}) + (\alpha_\lambda + \sigma_s)I_\lambda(\overrightarrow{r}, \overrightarrow{s})$$
$$= \alpha_\lambda n^2\frac{\sigma T^4}{\pi} + \frac{\sigma_s}{4\pi}\int_0^{4\pi} I_\lambda(\overrightarrow{r}, \overrightarrow{s}')\Phi(\overrightarrow{s} \cdot \overrightarrow{s}')d\Omega' \tag{8}$$

where $I_\lambda$ is the radiation intensity for wavelength $\lambda$, $\overrightarrow{r}$ is the solar position vector, $\overrightarrow{s}$ is the solar direction vector, $\alpha_\lambda$ is the spectral absorption coefficient, $\sigma_s$ is the scattering coefficient, $\sigma$ is the Stefan-Boltzmann constant, $n$ is the refractive index, $\Phi$ is the diffusion phase function, $\Omega'$ is the solid angle.

The steady-state simulation is performed to determine the temperature field in the unventilated state. The partial differential governing equations of the steady-state model are solved by

the pressure-velocity coupling algorithm. In the spatial discretization of the modelling, the least square cell based is adopted for the gradient, while the momentum and energy terms employ a second-order upwind scheme. Additionally, the discrete ordinates, the turbulent kinetic energy and dissipation rate of the realizable k-ε turbulence utilize the first order discretization scheme. The pseudo transient and warped-face gradient correction are active to improve the convergence ability of the simulation. Finally, the steady-state results are executed as initial values of the transient simulation to determine the greenhouse microclimate in the ventilated state, and the SIMPLE algorithm is applied to solve the governing equations. The residual monitors of the model are established at $10^{-6}$ for the energy equation and $10^{-3}$ for the continuity, turbulence and radiation equations. The computers are performed using parallel processing on a Z10PA-D8 server containing 24 Intel Xeon E5-2678 v3 2.50 GHz processors and 64GB Fully DDR4 Buffered memory.

**Computational domain and mesh layout.** A coupled computational dominant is illustrated in **Fig 3**. It is adopted to model the natural ventilation of arched greenhouses with continuous roof vents, in which the interaction between external wind-induced field and internal buoyancy-induced airflow is modelled simultaneously. The ground and plastic cover surface of the arched greenhouse are modelled the wall boundary conditions [38]. The lateral and top surfaces of the external flow field are set as the symmetry, while the external soil surface is set as a user-defined periodic temperature surface. The soil boundary under the greenhouse is set an adiabatic surface and does not consider participation in radiation due to the insulation structure for the thermal protection. In addition, the analysis object mainly focuses on the indoor microclimate, and the wall of the sliding cover is simplified into the heat insulation. The windward surface adopts the velocity inlet boundary condition. Moreover, a logarithmic streamwise velocity profile representing a neutral atmospheric boundary is imposed (**Eq 9**).

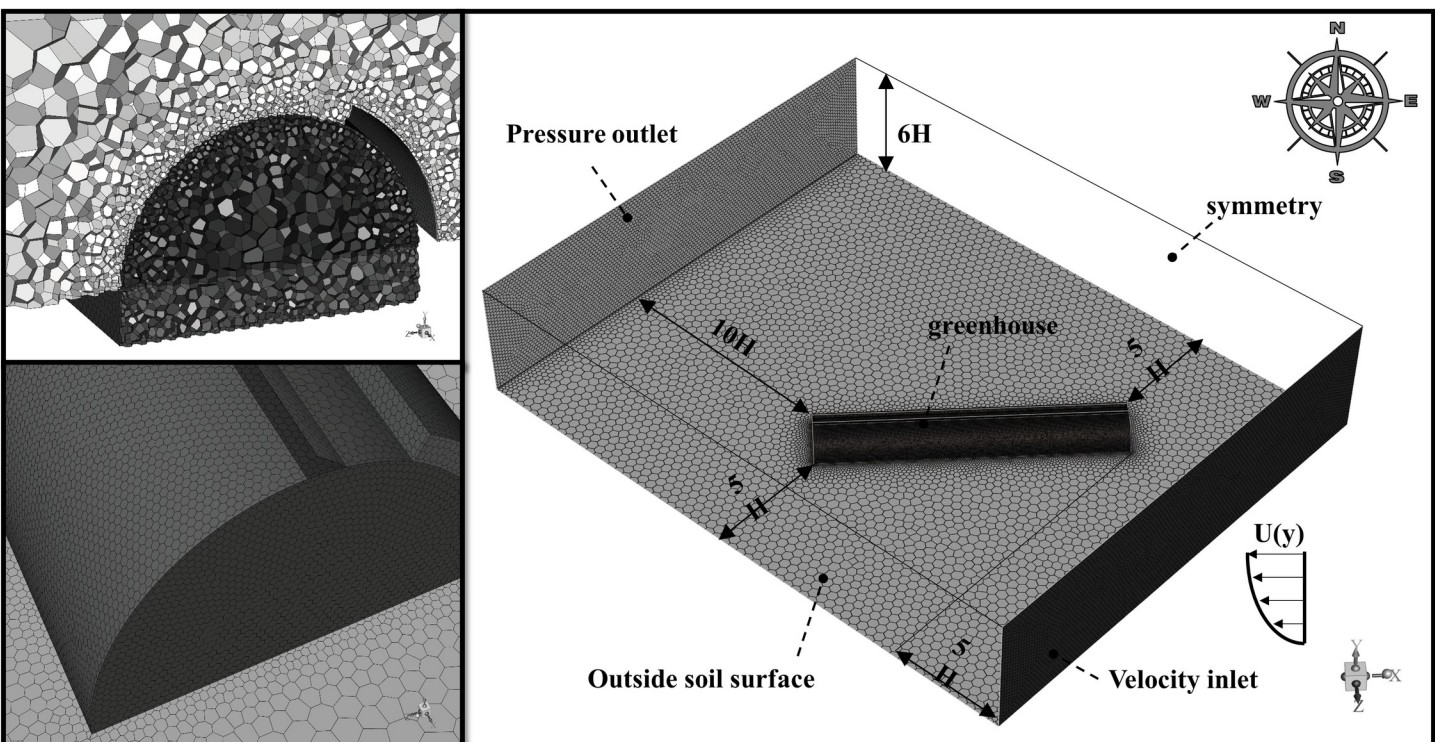

**Fig 3.** Detailed mesh structure (left) and the computational domain (right) of the arched greenhouse used for the numerical simulation.

Table 1. Thermophysical and optical properties of materials used in the simulation model.

| Property | Air | Soil | Polyolefin film |
|---|---|---|---|
| Density (kg m$^{-3}$) | 1.225 | 1700 | 950 |
| Thermal conductivity (W m$^{-1}$ K$^{-1}$) | 0.0242 | 0.85 | 0.19 |
| Specific heat capacity (J kg$^{-1}$ K$^{-1}$) | 1006.43 | 1010 | 1600 |
| Absorption coefficient | 0.1 | 0.5 | 0.15 |
| Refractive index | 1 | 1.92 | 1.7 |
| Emissivity | 0.86 | 0.9 | 0.85 |

The corresponding leeward surface is set as the pressure outlet. To avoid the potential influence of the reserved flow effect on the airflow in the greenhouse, the external flow field around the greenhouse should be kept at least 5H except the leeward side which is at least 15H away from the greenhouse, where H is defined as the ridge height of the arched greenhouse. In general, it is widely used to avoid the complex airflow analysis of the proximity of vent openings and accurately predict the greenhouse microclimate, following the procedure recommended in Lee et al. [39]. Thermophysical and optical properties of the materials used in the calculation domain are shown in **Table 1**.

$$\frac{U(y)}{U(y_r)} = \left(\frac{y}{y_r}\right)^{\zeta} \tag{9}$$

where $U(y)$ is the wind velocity at y height, $U(y_r)$ is the reference wind velocity at the reference height ($y_r = 10m$), $\zeta$ is the coefficient of roughness.

The geometry and meshes for the numerical analysis are designed using the Solidworks and Fluent Meshing software tools. During the gridding process, in order to reinforce the airflow pattern and thermal behavior in the greenhouses, and reduce the computational cost, the surface grids with a specified local sizing are created and then the computational geometry is discretized by a polyhedral mesh. The grid sensitivity analysis is performed based on three grids with different resolutions: a coarse grid with 388,596 cells, a middle grid with 623,242 cells and a fine grid with 1,257,554 cells. Detailed grid sizing and evaluation parameters are summarized in **Table 2**. The overall grid quality of different mesh sizes is evaluated by the grid discrete evaluation indicator and environmental parameter under the steady-state ventilation condition. The minimum orthogonal quality of the middle grid is 0.35 which is greater than the coarse grid and the fine grid 66.7% and 16.7%, respectively. The maximum skewness of the middle grid is 0.65 which is lower than the coarse grid and the fine grid 17.8% and 9.2%, respectively. Comparing the medium grid and the refined grid can be concluded that there is no significant discrepancy in average air temperature and velocity within 1 min of ventilation, but utilizing medium grid can save half the computational burden. Therefore, the steady-state simulations

Table 2. Comparison results of grid sensitivity analysis.

| Variables | Coarse grid | Middle grid | Fine grid |
|---|---|---|---|
| Number of elements | 388,596 | 623,242 | 1,257,554 |
| Volumetric region of the greenhouse | 111,974 | 204,496 | 469,909 |
| Minimum orthogonal quality | 0.21 | 0.35 | 0.3 |
| Maximum skewness | 0.79 | 0.65 | 0.71 |
| Discrete time | 2.28 min | 3.68 min | 7.6 min |
| Average air velocity | 0.43 m s$^{-1}$ | 0.43 m s$^{-1}$ | 0.44 m s$^{-1}$ |
| Average air temperature | 306.04 K | 305.79 K | 305.51 K |

are carried out and the middle grid is adopted in consideration of the calculation burden and the guarantee of simulation accuracy. Other contrastive cases in parameter investigation are meshed in the identical manner, and the total grid number is approximately 650,000.

**Characteristic paraments and roof ventilation structures.** In order to regular geometry parameters of the continuous roof vent, it is essential to have suitable variable indicators to satisfy the greenhouse design specifications in the natural ventilation process. The ventilation flow rate and microclimate inhomogeneity are regarded as the important characteristic parameters for identifying the ventilation performance [40]. The ventilation flow rate is identified as the number of the greenhouse air exchange per hour (**Eq 10**), which reflects the local ventilation rate in the arched greenhouse. To overcome the limitation of regional ventilation assessment, the microclimate inhomogeneity is proposed by **Eq 11**, which reflects the degree of local environmental variables and the overall deviation. In the simulation, the characteristic parameters can be obtained by calculating simulated the mass flow rate, air temperature and velocity.

$$VFR = 3600 \frac{\bar{m}_A A_V}{\rho V} \tag{10}$$

$$J_P = \frac{1}{P_M} \sqrt{\frac{\sum_{i=1}^{N} (P_M - P_A)^2}{N}} \tag{11}$$

where $VFR$ is the ventilation flow rate of the greenhouse, $\bar{m}_A$ is the average mass flow of the roof vent, $A_V$ is the area of the roof vent, $V$ is the greenhouse volume, $J_P$ is the microclimate inhomogeneity of the parameter (i.e. air temperature and velocity), $P_M$ and $P_A$ represent the mean value and simulated value of the climate parameters, respectively.

The prediction of ventilation performance is conducive to providing the detailed air circulation information in a ventilated greenhouse even before the greenhouse is constructed. Therefore, the parametric study of the roof ventilation configurations is performed considering the external conditions and greenhouse thermal microclimate. **Fig 4** displays the conventional geometry structures of the roof opening in the central section of the arched greenhouse. The performance of three ventilation structures of Rolling shutter type (RS), Jack-up type (JU) and Pivoting window type (PW) are evaluated. To illustrate the effects of different structures of the roof ventilation on the ventilation performance, the arc chord angle is designed at three different openings: 8°, 10°, 12°. Similarly, the position angle is designed at four different orientations: 75°, 85°, 95°, 105°. Combined with the actual production to ensure the rationality of the comparison analysis, the effective distance of the jack-up window type is set to 0.5 m, which corresponds to the pivoting window type flips 35° in the center of the semicircle. The

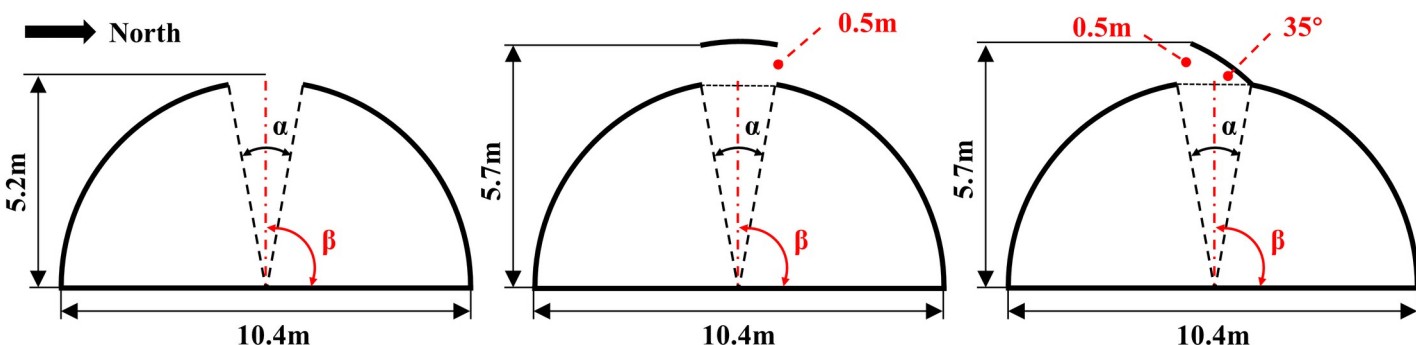

**Fig 4. Roof ventilated configurations of different ventilation structures of the arched greenhouse for CFD simulation.**

**Table 3. Analytical conditions for evaluation of natural ventilation using the CFD technology.**

| Classifications | Parameters |
|---|---|
| Ventilation structure | Rolling shutter type, Jack-up type and Pivoting window type |
| Arch chord angle (α) | 8˚, 10˚, 12˚ |
| Position angle (β) | 75˚, 85˚, 95˚, 105˚ |
| Wind direction at 10m height | 135˚ (optimistic ventilated structure: additionally, 90 and 180˚) |
| Wind speed at 10m height | 3.5 m s$^{-1}$ (optimistic ventilated structure: additionally, 1.5 m s$^{-1}$ and 5.5 m s$^{-1}$) |
| Water circulation heat collection system | 17˚C |
| Outside temperature | 24.5˚C |
| Outside soil surface temperature | 25.8˚C |
| Insolation clearness index | 0.4 |

analysis conditions of the parametric study using the steady-state simulation are summarized in **Table 3**.

## Results and discussion

### Validation of the simulation model

The purpose of the parametric study is to analyze the effect of the roof ventilation configuration in terms of natural ventilation, independently on the other potential microclimate factors such as the transpiration and photosynthesis, in particular crop planting layout. Moreover, The greenhouse proposed in this paper uses a water cycle heat collecting system to cool the greenhouse. Its main cooling principle is the thermal convection between air and water pipes, which is different from spray evaporation, and the transpiration of crops in the greenhouse is weak in the seedling stage. Therefore, a full-scale experiment has been conducted under the continuous roof ventilation condition, ignoring the impact of the evapotranspiration phenomenon on environmental benefits. To save the computing burden and enhance the experimental reproducibility, a three-dimensional steady-state CFD simulation is presented, which

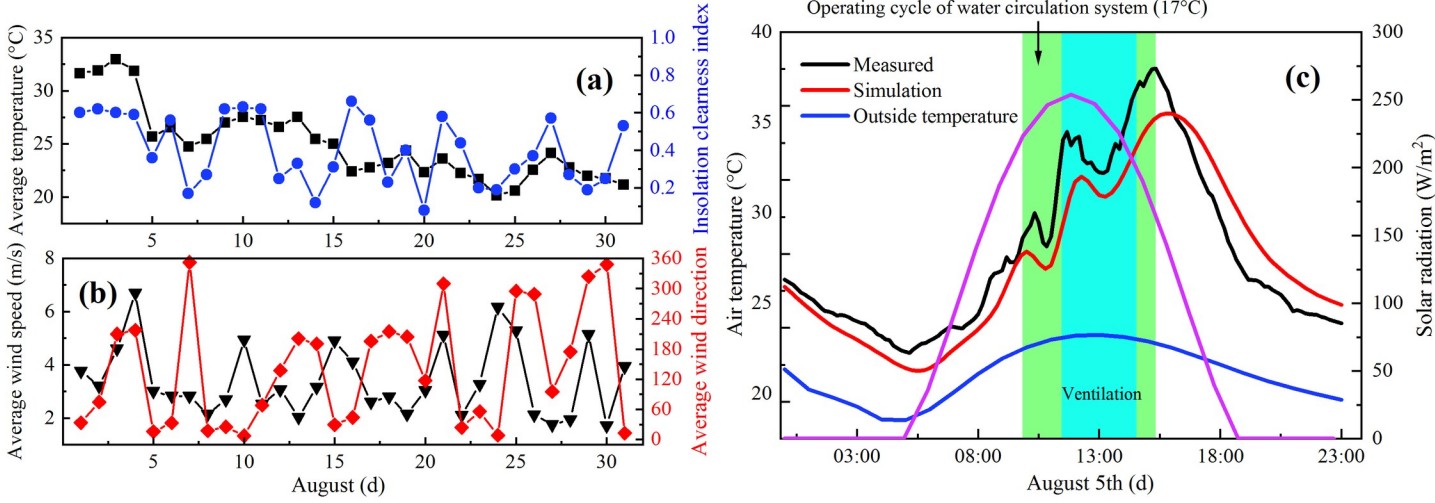

**Fig 5. Environmental parameters in August.** (a) Average air temperature and insolation clearness index of the outside; (b) Average wind speed and direction of the outside; (c) Thermal field of the ventilated greenhouse on August 5th for the model verification.

reproduced the thermal behavior of the arched greenhouses in the unventilated process. As shown in **Fig 5A and 5B**, the ambient parameters were derived from the monthly average data of August 2018. The monthly average data temperature and insolation clearness index are 24.5°C and 0.4, and the average wind speed and direction are 3.5 m s$^{-1}$ and 135°, respectively. To ensure the climate of the simulation model is consistent, all the average parameters are taken as the boundary conditions for the steady-state analysis model.

The reliability and accuracy of the CFD methodology need to be verified to make the relevant available data at satisfactory quality criteria. In this parameterization study, the theory of steady-state and transient-simulation models is consistent but the difference is in the time correlation. Therefore, the environmental parameters of the arched greenhouse on August 5th were recorded for the verification of the ventilation model, and the detailed records are displayed in **Fig 5C**. During the full-day trial, the operation time of the roof ventilation system was 11:30–15:00 to overcome the indoor high temperature environment. To quantify the agreement between simulated and measured values, the PRMSD and NMSE of the air temperature are calculated to be 5.69% and 9.95%, respectively. The experimental error of the airflow velocity in the roof vent is 18.5% owing to the unfavorable experimental operation. The assumption that skeletons and crops are ignored is an ideal simplification that may cause slight deviations between the full-scale measurements and simulated model, but does not affect the comparison of configuration scripts under the natural ventilation. In general, a good agreement between the simulation and measurements has been obtained, and the arched greenhouse model is used to perform the ventilation performance simulation, especially determine the optimal parameters of the arc chord angle and position angle of the roof vents.

## Analysis of airflow pattern and thermal behavior

Ventilation structures and vent positions of the arched greenhouse are designed in conjunction with the construction conditions of the greenhouse in northeast China. The prevailing summer wind direction in the northeast region is southeast wind and the simulated average wind speed is 3.5 m s$^{-1}$. The average air temperature and velocity at the time of ventilation 600s obtained from the simulation model are presented in **Fig 6**, which clearly shows that the roof vent structure of the arched greenhouse has a significant impact on the internal environment. In addition, it is observed that the average temperature vary inversely with the air velocity on the whole. However, the corresponding position angle of vent can affect the relationship

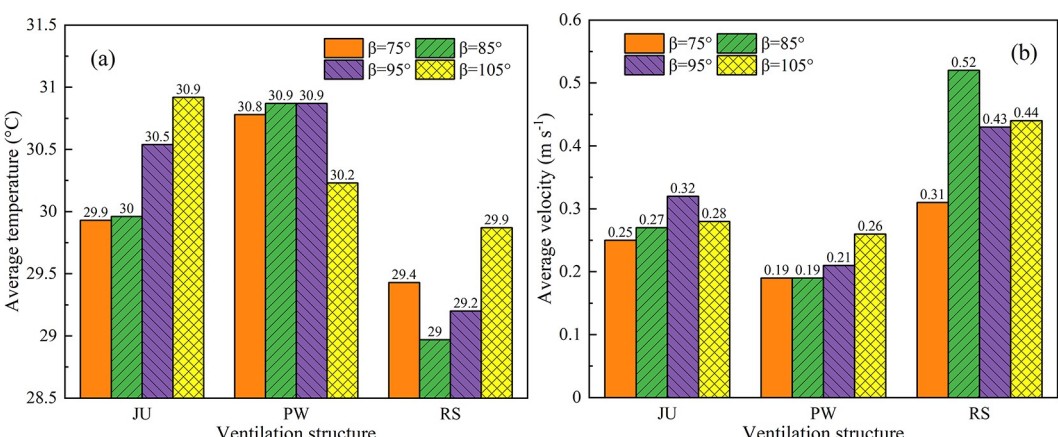

**Fig 6. Average air temperature and velocity of the arched greenhouses with different position angle and vent structure at the time of ventilation 600s.**

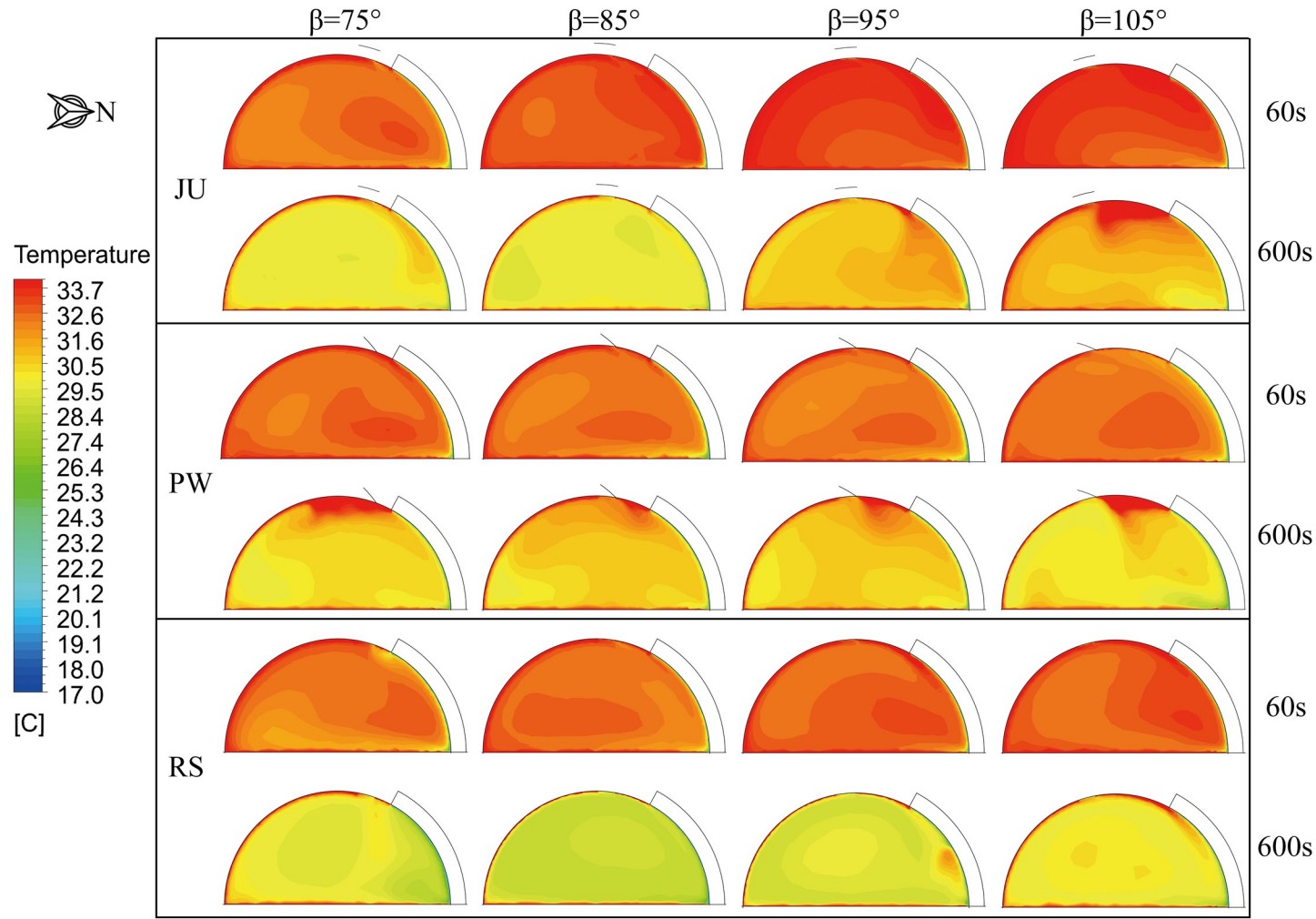

**Fig 7. Distribution of air temperature in vertical cross-section through the middle of the arched greenhouses.**

between the thermal behavior and airflow pattern due to the changes in the geometric characteristics of the roof ventilation structure. The fluctuation ranges of the overall average of greenhouse obtained at different position angles for air temperature and airspeed are 29.9–30.9°C and 0.25–0.32 m s$^{-1}$, 30.2–30.9°C and 0.19–0.26 m s$^{-1}$, 29–29.9°C and 0.31–0.52 m s$^{-1}$ for JU, PW and RS, respectively, and show that RS still has excellent cooling and ventilation performances under the condition of low internal airflow.

The simulated temperature distributions in the vertical cross-section through the middle of the arched greenhouses for the ventilation time 60s and 600s are illustrated in **Fig 7** when the outside wind speed is 3.5 m s$^{-1}$ and wind direction is 135°. Likewise, the airflow fields inside the greenhouse are closely related to the thermal behavior as shown in **Fig 8**. For the ventilation time of 60s, the temperature gradient in JU is obviously divided due to the roof vent is close to the water circulation system. The incoming airflow rotates clockwise along the north side of the greenhouse more favorably with the decrease of the position angle, which is conducive to the internal air heat exchange and ventilation airflow. The results show that the position angles of 75° and 85° in JU are efficient cooling schemes than that of 95° and 105° and the average temperature is reduced by 0.4°C. However, The position angle has a marginal effect

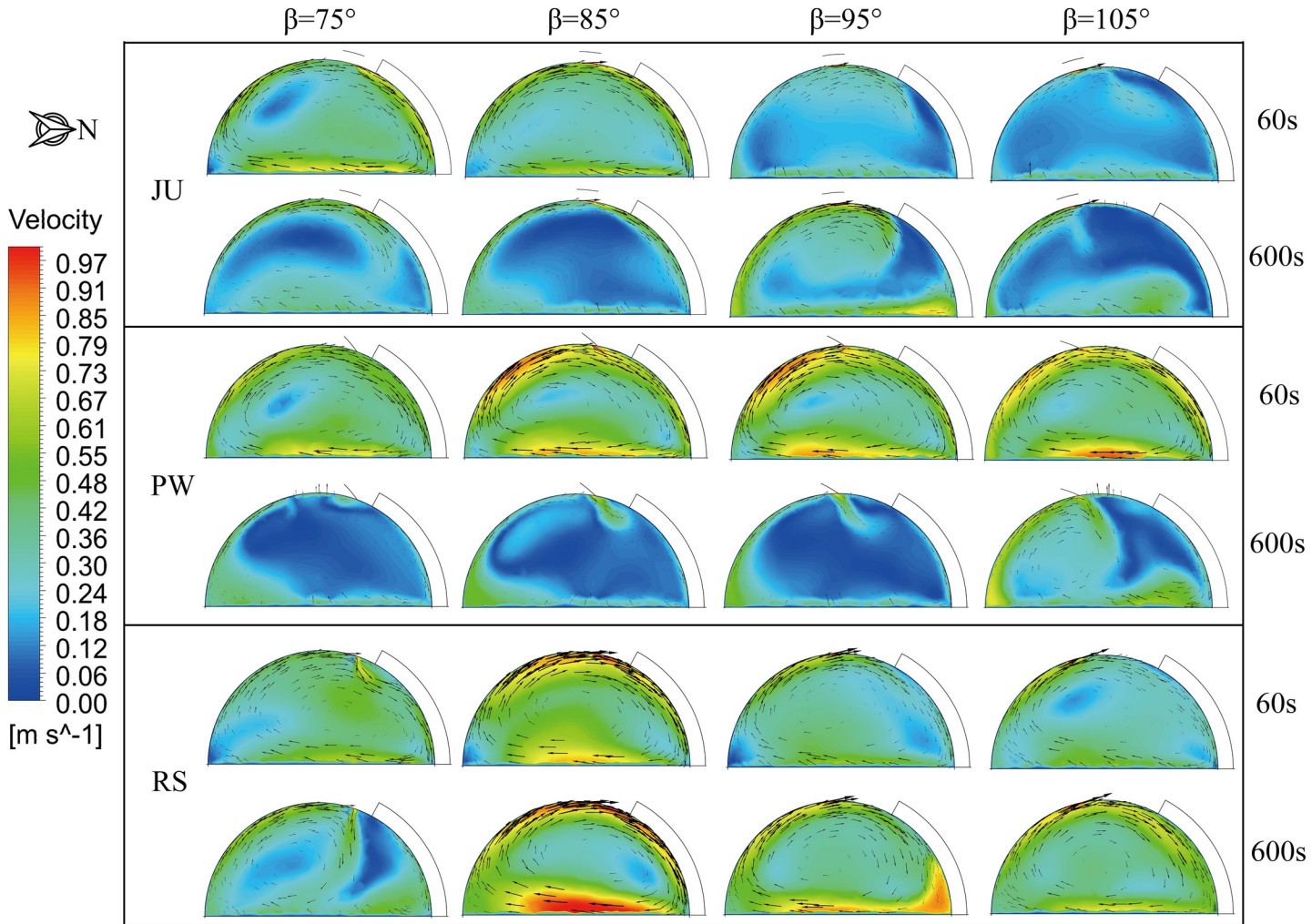

**Fig 8. Distribution of airflow field in vertical cross-section through the middle of the arched greenhouse.**

on the temperature distribution of PW, and the thermal mass is mainly located at the crop horizontal height 1.5 m on the north side of the greenhouse. The structural characteristics of the window flipping will lead to a certain guild effect on the external airflow patterns, resulting in a higher airflow velocity near the ground and under the south shed. The average air temperature of PW is basically reduced to 32.6°C at different position angles. Additionally, the renewal airflow from the roof ventilation will gradually transfer to the greenhouse floor as the position angle decreases for RS, hence, the position angle of 75° is satisfactory than 105° and the average temperature is reduced by 0.3°C.

For the ventilation time of 600s, the microclimate distributions of the arched greenhouse are maintained in a process of equilibrium ventilation. Under the same boundary conditions, the position angles of 75° and 85° have a maximum ventilation process for JU, which helps to alleviate the accumulation of overheated air at the top of the greenhouse. On the contrary, the thermal stagnation occurs near the roof vent of PW due to the forced action of the rotating window. The unfavorable structure causes the airflow in the central zone of the greenhouse to flow slowly, thereby resulting in the temperature at the top of the greenhouse is 2.5°C higher than the growing area. It is observed that the position angle of 85° is suitable for the pivoting

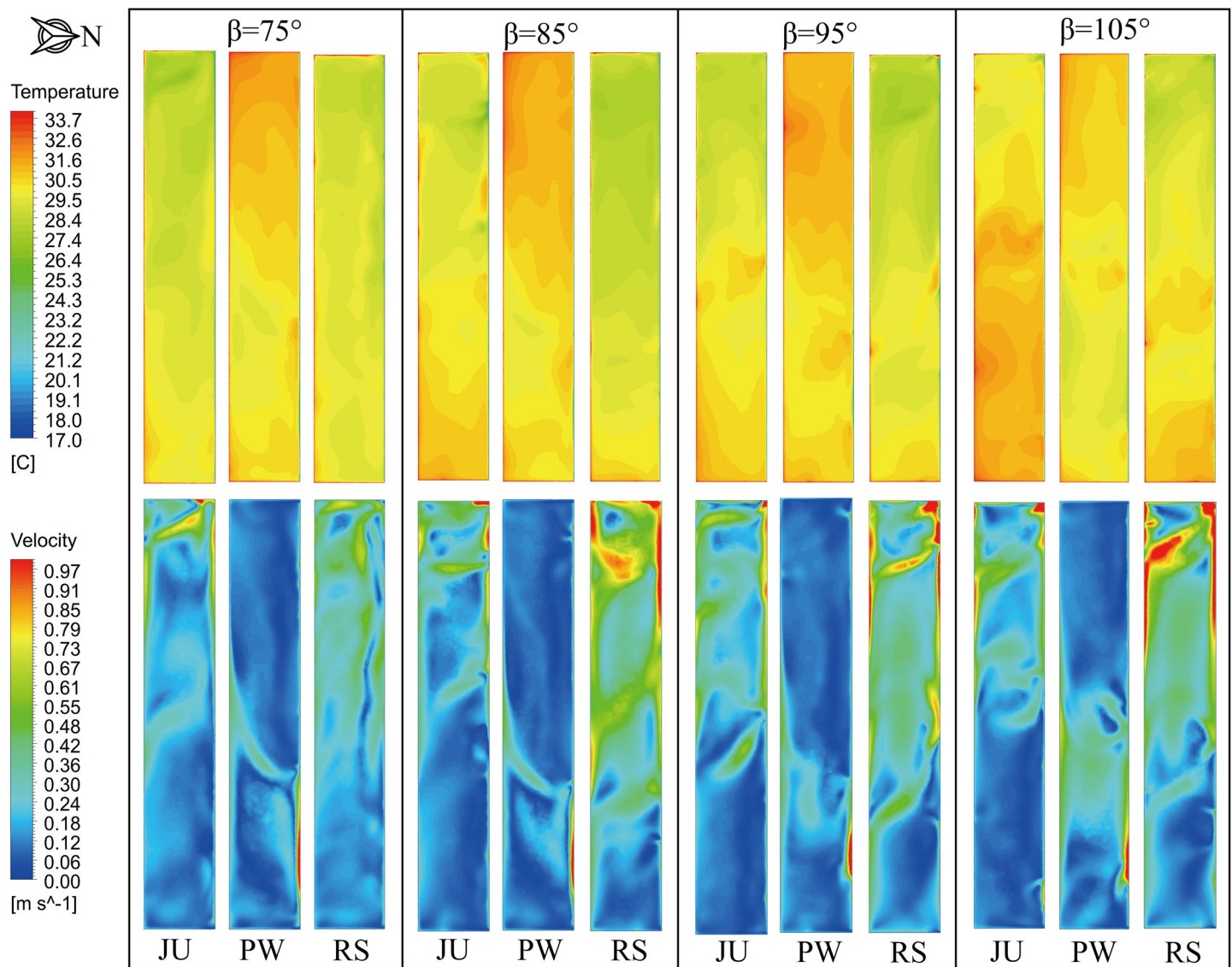

**Fig 9. Air temperature and velocity contours in a horizontal cross-section at a height of 1.5m for the arched greenhouses when the ventilation time is 600s.**

ventilation structure resulting in the internal average temperature drop of 0.6°C. When the position angle of RS is 75°, the eddy current leads to the deviation of the external airflow entering the roof vent to south roof, which is generated by the external sliding cover on the leeward side. Compared with other structures with the position angle of 85°, the RS with the position angle of 75° still has superior ventilation and cooling performances.

Fig 9 shows the air temperature and velocity distributions of the arched greenhouses at the 1.5 m horizontal cross-section when only the roof ventilator is fully open, wind speed is 3.5 m s$^{-1}$ and wind direction is 135°. The numerical simulation results show that the airflow velocity corresponding to the area with decent recirculation is higher, which can be inferred that the temperature distribution in the greenhouse is negatively correlated with the airflow distribution. On the other hand, the tendencies of the overall temperature gradient of PW is that the temperature in the west side is higher than that in the east side, which is exactly contrary to the results of JU and RS. Overlaid airflow is caused by the prevailing wind direction at the

ventilation intersection on the east side of the greenhouse and close to the north side. In general, it can be determined that designed JU and RS effectively improved the ventilation cooling process owing to the higher circular flow. Meanwhile, the position angle of 85˚ of the arched greenhouses is an optimum ventilation direction and its impact on the microclimate is marginally affected by the change of the ventilation structure. However, the airflow pattern and thermal behavior in the greenhouse is strongly related with the effect of the position angle. Comparing the scenarios RS, the climate characteristics of the position angle of 75˚ are remarkably uniform and stable. The results show that the position angles of 75˚in RS have similar cooling function than that of 85˚ under the condition of reducing the direct damage to crops influenced by the near-surface airflow. In a nutshell, the designed RS is recommended as the preferred option in the application of circular arch natural ventilation.

## Estimation of ventilation performances

In order to quantitatively assess the microclimate uniformity and cooling characteristics of RS at the position angle of 75˚, the ventilation flow rate and microclimate inhomogeneity are introduced to systematically evaluate the airflow pattern and thermal behavior in the arched greenhouse under different ventilation structures. **Fig 10** shows the histogram of the ventilation flow rate depending on the ventilation structure and position angle, for the case of a wind direction of 135˚ and a wind speed of 3.5m s$^{-1}$. The highest values of ventilation flow rate are found to occur at a position angle of 85˚ for the assumed ventilation forms. The total ventilation flow rate of RS and JU is 2.66 and 1.97 times higher than PW, respectively. This reflects that the rolling shutter type of roof vent can obviously enhance effective air exchange capacity

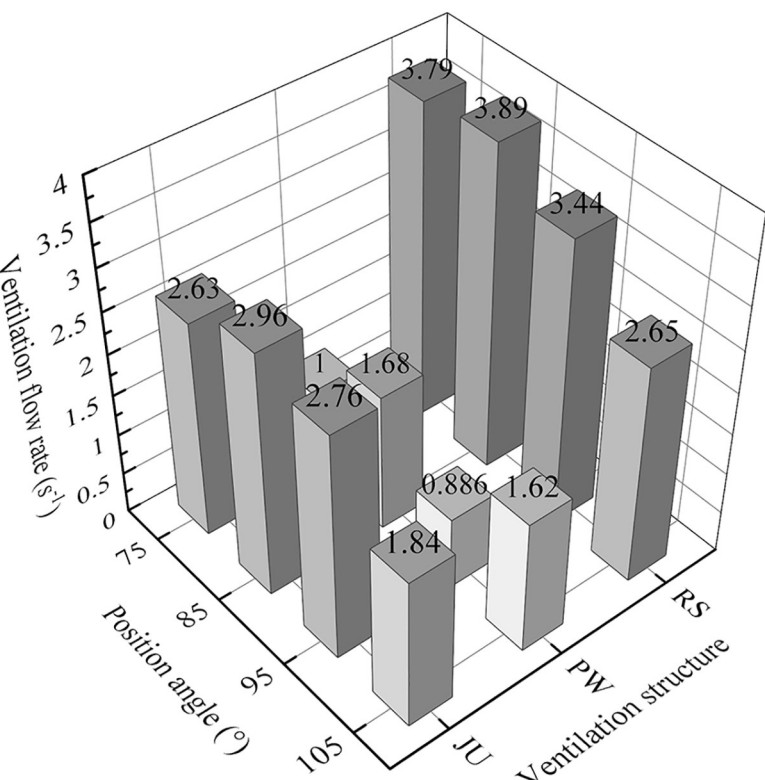

**Fig 10. Histogram of ventilation flow rate of the arched greenhouses with different position angle and vent structure.**

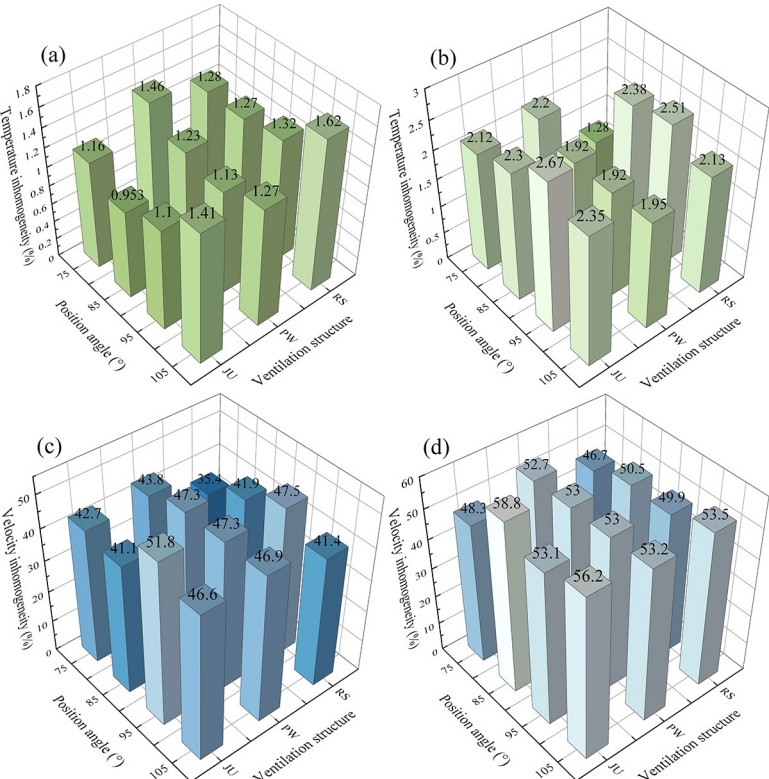

**Fig 11.** Histogram of microclimate inhomogeneity of the arched greenhouses with different position angle and vent structure at time steps of 60s (a, c), 600s (b, d).

in the arched greenhouse. Meanwhile, comparing the cases at different structure level, the values of flow rates at the position angles of 75˚ and 85˚ for RS are similar and they are completely superior to others.

To further analyze the microclimate distributions inside the arched greenhouse, the histogram of microclimate inhomogeneity of the temperature and velocity with different position angle and vent structure at time steps of 60s (a, c), 600s (b, d) are mentioned in **Fig 11**. The lower value of microclimate inhomogeneity reveals that the fluctuation amplitude of environment parameters is stable during the ventilation process. The temperature inhomogeneity (**Fig 11A**) and velocity inhomogeneity (**Fig 11C**) of PW at the position angle of 85˚ are 2.48% and 7.59% less than the average values in the designed scenarios, indicating that the microclimate characteristics of PW is relatively uniform at the time steps of 60s, but the cooling capacity is not particular superior. However, The temperature inhomogeneity (**Fig 11B**) and velocity inhomogeneity (**Fig 11D**) of RS at the position angle of 75˚ are 40.3% and 11.32% less than the average values in the designed scenarios. In addition, this simulated configuration has perfect air exchange capacity and cooling effect because the average air temperature can be reduced by 1.5˚C more than the existing greenhouse in 10 minutes of ventilation, and temperature and velocity inhomogeneities are approximately decreased by 33.3% and 11.89%, respectively. Therefore, the RS at the position angle of 75˚ is considered a significantly scheme to improve the overall ventilation process inside the arched greenhouse.

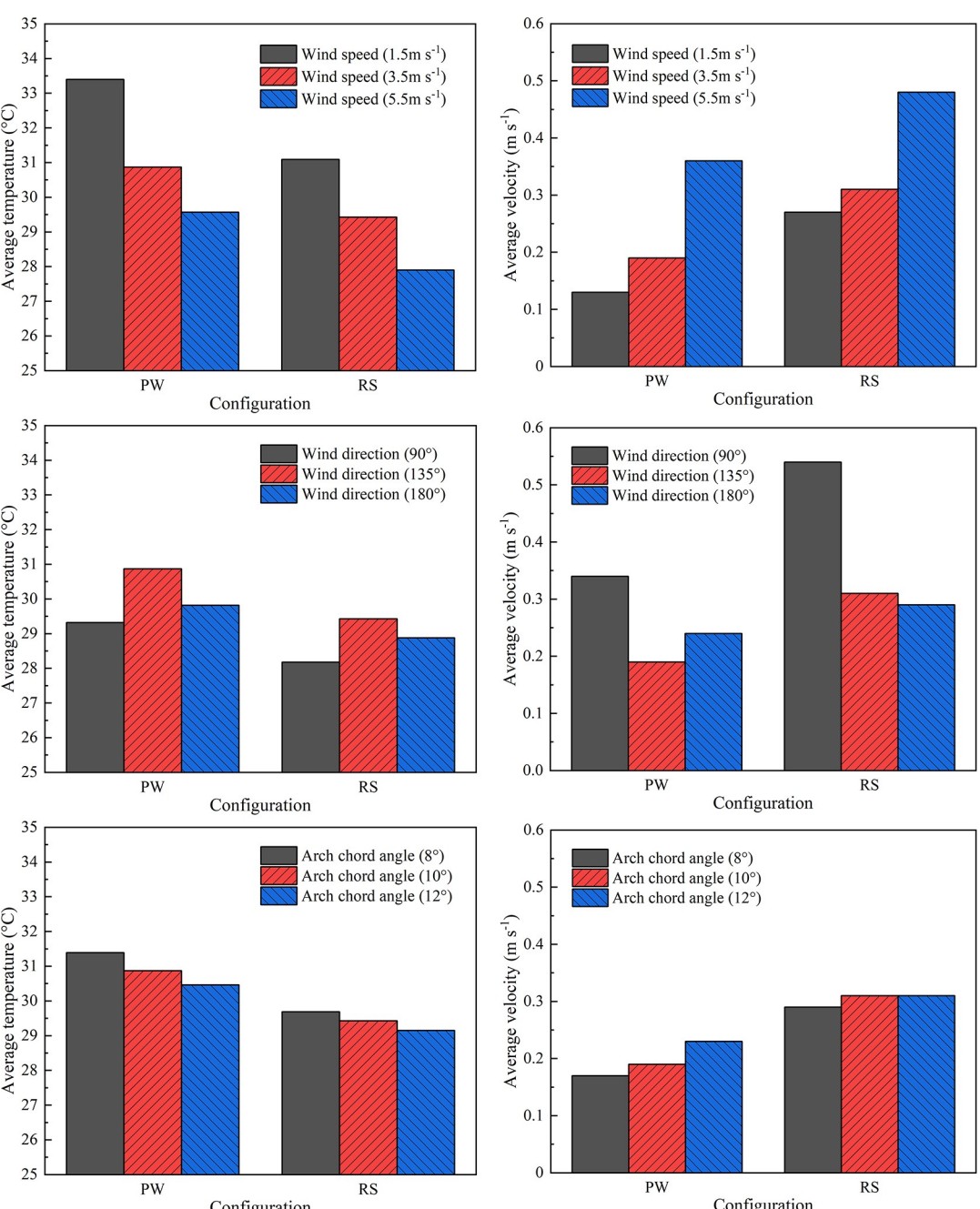

**Fig 12. Comparison of average temperature and velocity between experimental arched greenhouse and the simulated greenhouse with optimized ventilation structure under different wind speed, wind direction and ventilation opening conditions.** In all numerical models, the external wind speed defaults to 3.5m s$^{-1}$, the wind direction defaults to 135˚, and the arch chord angle defaults to 10˚.

## Assessment of microclimate performance

To evaluate the ventilation performance and average energy decrement of the designed RS with the position angle of 75˚, a quantitative comparison is performed to determine the average temperature and velocity in the arched greenhouse. In all numerical models, the external

wind speed defaults to 3.5m s$^{-1}$, the wind direction defaults to 135˚, and the arch chord angle defaults to 10˚. According to the simulation results as shown in **Fig 12**, only a single parameter is changed to explore the influence of wind condition and vent opening on the overall microclimate. Comparing the designed ventilation scenario (i.e. Rolling shutter type with the position angle of 75˚) and existing configuration (i.e. Pivoting window type with the position angle of 85˚), when the external wind speed rises from 1.5 m s$^{-1}$ to 5.5 m s$^{-1}$, the average temperature decreases from 33.4˚C to 29.57˚C for PW and 31.09˚C to 27.09˚C for RS, inversely, the average velocity increases form 0.13 m s$^{-1}$ to 0.36 m s$^{-1}$ for PW and 0.27 m s$^{-1}$ to 0.48 m s$^{-1}$ for RS. These results sufficiently justify the inverse relationship between the thermal behavior and airflow pattern, and further conclude that high wind speed can improve the cooling effect and increase the vortex flow rate of the greenhouse. In the case of ensuring the consistency of other climatic boundaries except the wind direction. The average air temperature of the greenhouses for the wind direction 90˚ and 180˚ is lower than 135˚, the difference is 1.55˚C and 1.05˚C for PW and 0.25˚C and 0.55˚C for RS, respectively. Meanwhile, the average velocity of the greenhouses for the wind direction 135˚ is higher than 90˚ and 180˚, with differences of 0.15 m s$^{-1}$ and 0.1 m s$^{-1}$ for PW and 0.23 m s$^{-1}$ and 0.25 m s$^{-1}$. The transform of prevailing winds reveals that the vertical blowing to the continuous ventilation opening of the greenhouse has a better cooling amplitude due to the excessive airflow velocity. To assess the effects of the roof opening on the microclimate features of the arched greenhouse, the arch chord angle is conducted based on economic benefits and ventilation configuration. It is observed from **Fig 12** that the influence of the arch chord angle on the temperature drop of the arched greenhouse is less than that of wind speed. The average temperature after 10 min of continuous roof ventilation is 31.39˚C, 30.87˚C and 30.46˚C for PW at the arch chord angle of 8˚, 10˚ and 12˚, respectively. The corresponding average temperature of RS is concluded as 29.69˚C, 29.43˚C and 29.15˚C.

Ventilation performance of the designed RS under the assumed conditions of wind speed, wind direction and arch chord angle has been compared in terms of ventilation flow rate and microclimate inhomogeneity in the arched greenhouse using the comprehensive numerical simulation (**Fig 13**). In all numerical models, the external wind speed defaults to 3.5m s$^{-1}$, the wind direction defaults to 135˚, and the arch chord angle defaults to 10˚.

For the wind speed as a single variable to compare the response performance, as the wind speed increased and the ventilation flow rate increased for PW, but the ventilation rate of RS decreases from 3.79 s$^{-1}$ to 3.13 s$^{-1}$ when the wind speed changes from 3.5 m s$^{-1}$ to 5.5m s$^{-1}$. The results show that RS is significantly superior to PW in terms of ventilation capacity because the average ventilation flow rate is 2.3 times higher at different wind speeds. The unfavorable microclimate inhomogeneity of the overall greenhouse of RS is observed under the condition of moderate wind speed (i.e. 1.5 m s$^{-1}$) because the weak turbulence benefit of the fresh air from the leeward side opening. For the wind direction as a single variable to compare the response performance, the ventilation flow rate is seriously insufficient for PW when the wind direction is 135˚, resulting in the worst cooling effect of natural ventilation. However, the temperature inhomogeneities for the proposed RS are 1.03%, 1.28% and 1.81% for the three assumed wind directions, and they are 0.51%, 0.69% and 1.85% lower than those with PW, helpful to avoid the influence of local high temperature on the crop development. The lowest velocity inhomogeneity of the proposed RS appears in the wind direction of 90˚, indicating that the prevailing wind blowing vertically to the ventilation opening to reduce the turbulent airflow. For the arch chord angle as a single variable to compare the response performance, it can be observed that the maximum ventilation flow rates of RS and PW occurred in the arch chord angle of 12˚ are 4.17 s$^{-1}$ and 2.54 s$^{-1}$, respectively. The ventilation flow rate and temperature inhomogeneity are inversely proportional to the increase of the arch chord angle. By

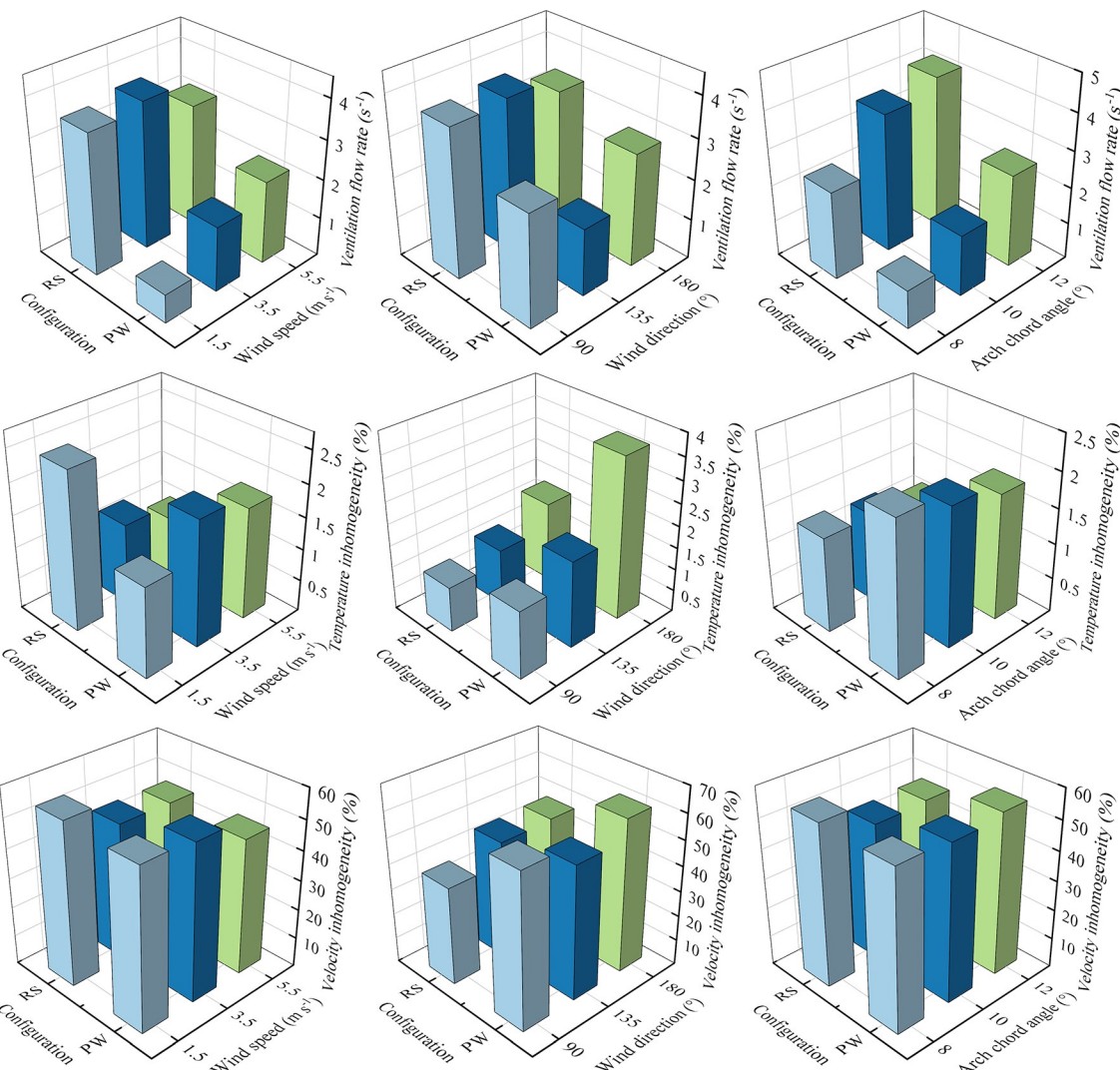

**Fig 13. Comparison of ventilation flow rate and microclimate inhomogeneity experimental arched greenhouse and the simulated greenhouse with optimized ventilation structure under different wind speed, wind direction and ventilation opening conditions.** In all numerical models, the external wind speed defaults to 3.5m s$^{-1}$, the wind direction defaults to 135°, and the arch chord angle defaults to 10°.

contrast, the velocity inhomogeneities of both have minimal fluctuation with the increase of the arch chord angle. In general, the proposed RS with the position angle of 75° is an effective cooling and ventilation strategy for the arch roof ventilation combined with the water circulation heat storage system.

## Conclusion

This paper presents a comprehensive numerical simulation to predict the airflow patterns and thermal behaviors in different arched greenhouses equipped with the water circulation heat storage system. A high-resolution polyhedral grid based on a grid sensitivity analysis is performed. The simulated model has been validated to ensure the credibility and reliability of simulated data. The ventilation flow rate and microclimate inhomogeneity are implemented to assess the natural ventilation performance under the roof vents of different position angle. Additionally, the

proposed RS with the position angle of 75˚ and the existing PW with the position angle of 85˚ are compared under different wind speed, wind direction and ventilation opening conditions.

The recommend ventilation structure is an effective cooling strategy to maintain the steady airflow and temperature fluctuations in the arched greenhouse. Moreover, the roof vent structure has a significant impact on the internal environment and the average temperature varies inversely with the air velocity on the whole. The results show that the corresponding position angle of vent can affect the relationship between the thermal behavior and airflow pattern due to the changes in the geometric characteristics of the roof ventilation structure. The position angle of 85˚ of the arched greenhouses is an optimum ventilation direction and its impact on the microclimate is marginally affected by the change of the ventilation structure. However, comparing the scenarios RS, the climate characteristics of the position angle of 75˚ are remarkably uniform and stable. The designed configuration has perfect air exchange capacity and cooling effect because the average air temperature can be reduced by 1.5˚C more than the existing greenhouse in 10 minutes of ventilation, and temperature and velocity inhomogeneities are approximately decreased by 33.3% and 11.89%, respectively.

To evaluate the ventilation performance and average energy decrement of the proposed RS with the position angle of 75˚, a quantitative comparison is performed to determine the average temperature and velocity in the arched greenhouse. The results sufficiently justify the inverse relationship between the thermal behavior and airflow pattern, and further conclude that high wind speed can improve the cooling effect and increase the vortex flow rate of the greenhouse. The transform of prevailing winds reveals that the vertical blowing to the continuous ventilation opening of the greenhouse has a better cooling amplitude due to the excessive airflow velocity. The average temperature after 10 min of continuous roof ventilation is 31.39˚C, 30.87˚C and 30.46˚C for PW at the arch chord angle of 8˚, 10˚ and 12˚, respectively. The corresponding average temperature of RS is concluded as 29.69˚C, 29.43˚C and 29.15˚C. Furthermore, the temperature inhomogeneities for the proposed RS are 1.03%, 1.28% and 1.81% for the three assumed wind directions, and they are 0.51%, 0.69% and 1.85% lower than those with PW, helpful to avoid the influence of local high temperature on crop development.

The practical value of the research is supposed to provide the basic quantitative conclusions for evaluating the airflow pattern and thermal behavior via natural ventilation of the arched greenhouses. The influence of crop growth conditions on the ventilation configuration is ignored due to the airflow resistance exhibited by the crop depends on the crop type and growth stage. Likewise, The impact of crops cannot qualitatively change the ventilation mechanism. More effective and intelligent research combining crop growth patterns and microclimate characteristics will focus on future work.

## Supporting information

**S1 Nomenclature.**
(DOCX)

## Acknowledgments

The authors are grateful to Horticulture Facility Design & Environmental Control Research Institute for supporting the project.

## Author Contributions

**Conceptualization:** He Li, Subo Tian, Tianlai Li.

**Data curation:** Yiming Li, Xingan Liu.

**Formal analysis:** He Li.

**Funding acquisition:** Xingan Liu.

**Investigation:** Xingan Liu, Subo Tian, Tianlai Li.

**Methodology:** He Li, Yiming Li.

**Project administration:** Xiang Yue, Xingan Liu, Subo Tian.

**Resources:** Xiang Yue, Xingan Liu, Subo Tian, Tianlai Li.

**Software:** He Li, Yiming Li.

**Supervision:** Xiang Yue, Subo Tian.

**Validation:** Yiming Li, Xiang Yue.

**Visualization:** He Li.

**Writing – original draft:** He Li.

**Writing – review & editing:** He Li, Tianlai Li.

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
