## [Decision Letter · Decision Letter 0]

8 Sep 2020

PONE-D-20-21515

Evaluation of airflow pattern and thermal behavior of the arched greenhouses with designed roof ventilation scenarios using CFD simulation

PLOS ONE

Dear Dr. Liu,

Thank you for submitting your manuscript to PLOS ONE. After careful consideration, we feel that it has merit but does not fully meet PLOS ONE’s publication criteria as it currently stands. Therefore, we invite you to submit a revised version of the manuscript that addresses the points raised during the review process.

We look forward to receiving your revised manuscript.

Kind regards,

Djamel Eddine Ameziani, Prof

Academic Editor

PLOS ONE

Journal Requirements:

Additional Editor Comments (if provided):

Dear Xingan Liu

Thank you for the manuscript which you have submitted for possible publication in the PLOS One Journal. I am pleased to advise you that the recommendations from two reviewers in the field are positive. I have also reviewed the paper and find it to be of good quality. Based of these evaluations, I am pleased to advise you that I intend to accept your work for publication with minor remarks (Those of reviewers).

Sincerely yours

Reviewers' comments:

Reviewer's Responses to Questions

**Comments to the Author**

1. Is the manuscript technically sound, and do the data support the conclusions?

Reviewer #1: Yes

Reviewer #2: Yes

2. Has the statistical analysis been performed appropriately and rigorously? 

Reviewer #1: Yes

Reviewer #2: Yes

3. Have the authors made all data underlying the findings in their manuscript fully available?

Reviewer #1: No

Reviewer #2: Yes

4. Is the manuscript presented in an intelligible fashion and written in standard English?

Reviewer #1: Yes

Reviewer #2: Yes

5. Review Comments to the Author

Reviewer #1: After I revised the paper, I have the following comments:

- The Nomenclature was not provided.

- The English should be revised. For example, in the lines 183 and 186, “and” should be added.

- In modeling and numerical simulation section, a commercial software was used to accomplish the simulations but it is not mentioned.

- It has been concluded from a lot of studies and researches that the evapotranspiration phenomenon have an important influence on the greenhouse microclimate, but it was ignored in this study. You should mentioned why this phenomenon was ignored in the current greenhouse modeling and simulation.

Reviewer #2: Interesting paper and the subject treated falls in the scope of the Journal, it has been carefully reviewed and does not requires any modification. So the reviewer suggests the acceptance of this manuscript for publication.

6. PLOS authors have the option to publish the peer review history of their article (what does this mean?). If published, this will include your full peer review and any attached files.

Reviewer #1: **Yes: **Choab Noureddine

Reviewer #2: No

---

## [Author Response · Author response to Decision Letter 0]

11 Sep 2020

Dear editor and reviewers:

The authors would like to acknowledge the encouraging and valuable comments concerning our manuscript entitled “Evaluation of airflow pattern and thermal behavior of the arched greenhouses with designed roof ventilation scenarios using CFD simulation”. (PONE-D-20-21515).Those comments are very helpful for revising and improving our paper, as well as the important guiding significance to our researches. We have studied the comments carefully and made corrections which we hope meet with approval. The main corrections are in the manuscript and the responds to the reviewers’ comments are as follows.

Replies to the reviewers’ comments:

Reviewer #1:

Response: Authors would like to thank the reviewer for his/her precious time spent in the revision of this paper. Considering the Reviewer’s suggestions, we tried our best to improve the manuscript. All of the inappropriate clarifications have been modified in the revised manuscript.

1.The Nomenclature was not provided.

Response: Thanks, we agree. The nomenclature have been provided. In the re-submitted manuscript, the abbreviations that appear in tables have been added. The above explanation has been included in the revised manuscript (see line 33).

2.The English should be revised. For example, in the lines 183 and 186, “and” should be added.

Response: The authors agree that the original statement is inappropriate. The statement has been revised in the revised manuscript (see line 186 and 193)

3.In modeling and numerical simulation section, a commercial software was used to accomplish the simulations but it is not mentioned.

Response: This paper presents detailed three-dimensional simulations of natural ventilation in arched greenhouses in relation to the designed roof ventilation scenarios with ANSYS 19.2 commercial CFD software. In addition, The geometry and meshes for the numerical analysis are designed using the Solidworks and Fluent Meshing software tools. The above explanation has been included in the revised manuscript (see line 186-190).

4.It has been concluded from a lot of studies and researches that the evapotranspiration phenomenon have an important influence on the greenhouse microclimate, but it was ignored in this study. You should mentioned why this phenomenon was ignored in the current greenhouse modeling and simulation.

Response: Thanks so much for your suggestion. Several researchers have systematically investigated the evapotranspiration phenomenon of different greenhouse. However, the purpose of the parametric study is to analyze the effect of the roof ventilation configuration in terms of natural ventilation, independently on the other potential microclimate factors such as the transpiration and photosynthesis. Moreover, The greenhouse proposed in this paper uses a water cycle heat collecting system to cool the greenhouse. Its main cooling principle is the thermal convection between air and water pipes, which is different from spray evaporation, and the transpiration of crops in the greenhouse is weak in the seedling stage. Therefore, a full-scale experiment has been conducted under the continuous roof ventilation condition, ignoring the impact of the evapotranspiration phenomenon on environmental benefits. The above explanation has been included in the revised manuscript (see line 342-351).

Reviewer #2: Interesting paper and the subject treated falls in the scope of the Journal, it has been carefully reviewed and does not requires any modification. So the reviewer suggests the acceptance of this manuscript for publication.

Response: Thanks very much for your kind work and consideration of the publication of our paper. On behalf of my co-authors, we would like to express our great appreciation to the reviewer.

Once again, thank you very much for your constructive comments and suggestions which would help us both in English and in depth to improve the quality of the paper.

Kind regards,

He Li

---

## [Editor Report · Decision Letter 1]

15 Sep 2020

Evaluation of airflow pattern and thermal behavior of the arched greenhouses with designed roof ventilation scenarios using CFD simulation

PONE-D-20-21515R1

Dear Dr. Liu,

We’re pleased to inform you that your manuscript has been judged scientifically suitable for publication and will be formally accepted for publication once it meets all outstanding technical requirements.

Kind regards,

Djamel Eddine Ameziani, Prof

Academic Editor

PLOS ONE
---

## [Editor Report · Acceptance letter]

17 Sep 2020

PONE-D-20-21515R1 

Evaluation of airflow pattern and thermal behavior of the arched greenhouses with designed roof ventilation scenarios using CFD simulation 

Dear Dr. Liu:

I'm pleased to inform you that your manuscript has been deemed suitable for publication in PLOS ONE. Congratulations! Your manuscript is now with our production department. 

Kind regards, 

on behalf of

Dr. Djamel Eddine Ameziani 

Academic Editor

PLOS ONE